# Social Capital and Rural Health for Refugee Communities in Australia

**DOI:** 10.3390/ijerph20032378

**Published:** 2023-01-29

**Authors:** Lillian Mwanri, Emily Miller, Moira Walsh, Melanie Baak, Anna Ziersch

**Affiliations:** 1Centre for Public Health Research, Equity and Human Flourishing, Torrens University Australia, Adelaide Campus, Adelaide, SA 5000, Australia; 2College of Medicine and Public Health, Flinders University, Bedford Park, SA 5042, Australia; 3UniSA Education Futures, University of South Australia, Adelaide, SA 5072, Australia

**Keywords:** refugee, resettlement, social capital, social determinants of health, rural settings

## Abstract

Refugee resettlement significantly contributes to Australia’s migration programs, with recent policy directives prioritising rural resettlement. As a result, the cultural diversity of populations of several Australian rural towns has substantially expanded. Newcomers may encounter challenges becoming part of closed social networks and accessing the resources they need for a healthy life in resettlement. However, there are also benefits that stem from positive integration for newcomers and for receiving populations. As part of a larger study, which aimed to explore facilitators and barriers to successful resettlement in a rural setting, the objective of this paper was to show how social ties were important for participants’ health, both facilitating access to resources, including health services, and connecting people to health-promoting living conditions. In-depth individual interviews with 44 participants from refugee communities originally from Africa and South-East Asia, settled in a rural South Australian town, were conducted. Participants were invited to the study through snowball sampling via known connections between the researchers and key people within the communities. Interview transcripts were analysed using framework thematic analysis. The findings demonstrate how participants drew on connections within their cultural communities, reflecting collectivist cultural values. These social ties were key to enabling access to resources for health. These included emotional resources, such as a sense of belonging, as well as physical and practical resources, including food, housing, and/or accessing services. Several participants were also working towards a career in the health industry. Populations from refugee communities in rural towns are growing, not only with the continuation of new arrivals to these towns, but also as the settled populations expand their families and communities. Effective health service provision in these locations needs to serve these growing communities, and there is scope for services to tap into community networks to assist with this.

## 1. Introduction

Globally, over 103 million people are currently forcibly displaced from their homes [1], with around one third officially recognised as refugees or asylum seekers. Dispersal policies in many countries aim to spread the settlement of people from refugee and asylum-seeker’s communities, aiming to revitalise rural economies [2], avoid perceived segregation issues associated with the settlement of ethnic groups in metropolitan settings [3] and provide a welcoming place for people to settle and make social connections [4]. Australia is one of the most multicultural nations, and the number of migrant populations has increased significantly, reported to be higher than in most Organisation for Economic Co-operation and Development (OECD) countries [5]. In 2020, nearly 30% of Australians were born overseas and a further 20% had at least one parent born overseas [6]. Australia has also committed to the resettlement of over 12,000 new refugees, and the net overseas migration contributes to over 60% of Australia’s total population growth. In Australia, resettlement of refugees in rural areas shapes population and economic growth as well as cultural and linguistic diversity in these locations [7,8,9,10]. As such, studies that provide evidence to inform policies and frameworks for practices that promote successful resettlement are a necessity.

Health and wellbeing are key elements of integration and resettlement [11]. The social determinants of health (SDH) lens offers a framework to consider the social, economic and cultural factors that impact health, and the influence of effective healthcare provision and access [12,13]. Through the analysis of interview data from 44 participants from refugee communities in rural South Australia, this paper seeks to show the ways in which social ties were important for participants’ health. The findings show that social capital was critical for health and wellbeing, enabling access to resources for health, including a sense of belonging and welcome, housing, employment, and healthcare. In this article, we therefore discuss participants’ social ties and their impact on the SDH, and the potential for service provision that connects with community networks for improved health outcomes.

### 1.1. Refugee Resettlement in Rural Areas: The Australian Context

Australia has a colonial history that affects contemporary in/exclusion for those from Indigenous, migrant or refugee communities [14]. Populist and political rhetoric often display exclusionary racialised discourses [15], which constitute the ways that people from migrant or refugee communities are portrayed and perceived [16]. There are health, education, employment, and incarceration figures that show there is work to be performed towards (multi)cultural inclusion and equitable outcomes [17,18,19]. Despite these challenges, Australian society is generally considered multicultural, with almost a third of the Australian population being born overseas, around ten per cent of whom arrive in Australia annually via the Refugee and Humanitarian Program [20].

Australian refugee resettlement generally involves government support during initial settlement, including English language lessons, temporary housing upon arrival, and access to specialised health services [21], though this is dependent on visa type, with eligibility restricted for temporary visa holders such as asylum-seekers. Over time, people are expected to access non-specialised services (usually within a year but can be extended in cases of complex needs). There is low cost or free access to government schooling and healthcare, and income support, available to all permanent residents or citizens. In recent years, there has been an emphasis on humanitarian resettlement in regional locations, including small towns often surrounded by agricultural land [22], which we herein refer to as ‘rural’.

Rural Australian life presents both opportunities and challenges; people from migrant or refugee communities may find a sense of safety and some opportunities for employment but limited access to services or adequate upward mobility [23,24]. Contributions to social connectedness have been noted at the individual, community, institutional and governmental level [4,22,25,26]. However, although some country towns are inclusive and welcoming places for new cultural communities to settle, in other cases racialised discrimination can preclude this connection to the wider community and equitable access to goods or services [[27]，[28]]. In these cases, relationships within cultural communities can be particularly important [29].

### 1.2. Theoretical Frameworks

#### Social Capital, Integration, and the Social Determinants of Health

Social capital has been extensively debated but broadly refers to the resources available through social networks [30]. Key social capital theorists have discussed the role of trust, relationships, reciprocity, and networks at the individual and community level [31,32,33,34]. In relation to settlement experiences of people from refugee communities, social capital theories have been drawn on to examine the ways that social ties facilitate access to resources [11,35,36,37,38,39,40,41,42,43,44,45] including in rural contexts [11,23,46,47,48]

We draw on a conceptualisation of social capital, reflecting the work of Bourdieu [49], as the resources available through social networks [30]. This definition focuses both on the infrastructure of social capital (the social ties a person may have) as well as the resources, and their relative value, that this infrastructure may give rise to [30]. Capital can be used to lever and negotiate positive outcomes, but capital that holds value in one setting may not be relevant to another [31]. The relative value of different types of ties can depend on the social ‘field’ in which they are operating [49], for example, the ‘field’ of a specific locality, such as a rural area. Social relationships, or ‘ties’, have been differentiated as bonding, bridging, and linking [50]. While our perspective of social capital differs from Putnam’s focus on community level social capital, we do draw these characterisations of ties. Bonding ties are considered as close relationships within a group or network within which people are connected by shared ideals or cultural norms and practices, such as a family. Bridging ties are those that people create to other networks, to dissimilar others, and linking ties are vertically oriented (i.e., made between individuals where a clear power differential exists or between community organisations/individuals and institutions/government departments).

The notion of a distinct dichotomy between bonding and bridging ties has, however, been debated. Scholars have suggested that these relationships or ties occur on a continuum and that it is the way that access to resources are facilitated via relationships that is most relevant rather than categorising those relationships as only bonding or bridging [51,52]. Power relations between people in different networks or contexts are also crucial and can enable or block access to resources. The resource access supported within a particular social network (such as a co-ethnic cultural community) might therefore have limited reach if other social networks are closed—this is a key component of Bourdieu’s conceptualisation of social capital, as he highlights the way that social capital can perpetuate existing inequities [53]. In other words, a network of bonding ties does not necessarily facilitate access to resources and is not synonymous with social capital, as the access to resources is the capital [54]. We therefore use the terms bonding and bridging to help frame our discussion, but note that ties are not simply one or the other but exist on a continuum, which is impacted by power relations, access to other forms of capital, social location, and temporal factors.

In a resettlement context, it is relevant to consider the nature of bridging ties between those migrating and people from a majority settlement society, and the relative access to resources that bridging ties may facilitate. However, recent work has noted that in some places superdiversity means that there may not be an identifiable majority population with which newcomers can form bridging ties [55]. In addition, notions of bonded networks within cultural communities of people producing segregation in resettlement have also been questioned, and some research has highlighted strengths within cultural community networks, showing how access to resources is facilitated within communities [56], which are a source of cultural wealth [57]. A part of this cultural wealth is that trust and reciprocity within the community provides access to the resource of belonging, ‘intimate relationships, companionship, self-esteem and purpose’ [55]. However, there remain barriers to accessing certain resources for newcomers that bonded networks within cultural communities may not be able to assist with. Bridging and linking ties with individuals who can facilitate access to resources then become important, such as through faith groups, or through key brokers within a cultural community who can act to bridge or link community members with services, institutions or government departments [56].

Social capital is a useful lens through which to consider how integration is affected by the development of relationships between newcomers and receiving communities, a process that occurs in the context of historical and contemporary policies, practices and attitudes [58]. Integration is a theoretical construct used to discuss bi-directional adaptations that occur or are needed when people enter a new context after migration and highlights the importance of social connections and networks for accessing resources such as housing, education, employment, and health [11].

A consideration of social capital in rural settings is important not just in understanding the integration process but also because of the established links between social capital and health, given that social capital is an important SDH [59]. Social Determinants of Health are the ‘conditions in which people are born, grow, work, live, and age...[which] are shaped by the distribution of money, power and resources at global, national and local levels’ [13,60]. The SDH framework can help to outline the individual lifestyle factors (e.g., exercise, smoking), social and community networks, living and working conditions (e.g., education, work environment, housing) and general socioeconomic, cultural, and environmental conditions that affect health [12]. In the case of social capital and health for refugees in rural settings, the resources available through networks can be of assistance in providing emotional and practical support important for health, as well as facilitating access to other social determinants of health such as employment and housing and in encouraging healthy behaviours as well as accessing health services [61,62,63]. Such networks are particularly important in rural settings, given often greater resources constraints.

Elements of integration overlap considerably with the SDH and therefore both frameworks are useful for considering the social factors beyond individual behaviours that impact the health of people who migrate, either voluntarily or after a refugee experience [59]. In this way, social capital is a crucial element of both integration and the SDH, and a theoretical approach that considers these interrelated elements is relevant when considering the ways in which health can be achieved after refugee resettlement.

### 1.3. Rural Health and Resettlement

Globally, rural health and healthcare access are generally poorer compared to those in urban contexts, partially due to factors impacting health in these locations alongside difficulties regarding healthcare provision and access [64]. The relative incidence of injuries and mental health challenges are generally higher in rural locations [65,66]. Access to healthcare can be challenged by distances that increase time and expense related to travel, affecting both healthcare practitioners and consumers [67]. In Australia, these distances can be vast and rural health services are often affected by worker shortages [64,68]. Health outcomes are also affected by power relations and social structures, in addition to geographical challenges, in rural locales [69].

For people from refugee communities, the impact of these factors on health can be heightened. The barriers to effective healthcare provision facing former refugees settling in Australia have been noted in urban contexts, where the healthcare and settlement systems require adjustment to effectively address the needs of resettling refugees [59]. The few studies of these experiences in rural areas highlight additional challenges [70,71], with resettling refugees’ lack of understanding of often opaque Australian systems combining with a lack of appropriate cultural and linguistic support from healthcare services and practitioners [72]. Although some healthcare workers can develop culturally sensitive approaches to providing healthcare for these populations, it takes time to develop these skills and there can be a loss of these staff due to the general challenges and turnover in these often remote locations [73].

Wakerman and colleagues’ review of healthcare provision in rural and remote Australia [74] showed that a range of approaches to healthcare provision enabled responses to different needs of often small populations with limited infrastructure and resources. Their framework highlighted that essential service requirements for small rural towns, such as the one in our study, include linkages between health services and community, as well as a focus on workforce recruitment and retention. This article addresses these two requirements of rural healthcare provision, by drawing on a social capital and SDH framework to understand the ways in which resources for health were accessed by study participants.

We report the findings from our qualitative study that examine how social ties were important for refugee health, both in terms of facilitating access to resources, including health services, and connecting people to health-promoting living conditions that are SDH. We also identify potential opportunities for healthcare provision that builds on and connects to community and social networks.

## 2. Materials and Methods

This study comprised semi-structured interviews with 44 participants from refugee communities originally from two distinct regions: Africa and South-East Asia. Data were collected during 2016 in a small town of less than 30,000 people in rural town in South Australia. The town was located around four or five hours from the two closest cities of Melbourne and Adelaide (by car). This town has been a focus for dedicated humanitarian rural resettlement (known as ‘regional resettlement’ in the Australian vernacular) since 2007.

Participants were invited to the study through snowball sampling that began via previous connections between the researchers and key people within the communities. The interviews were conducted in people’s homes or a local hotel room, with interpreters made available when relevant and as negotiated with individual participants. Interviews were recorded and English sections transcribed prior to analysis, which was performed with the assistance of Nvivo software. The Nvivo assisted with coding and analysis of the qualitative data (transcripts from participants’ interviews). Framework analysis was utilised, which involved familiarisation, development and application of a coding framework, and charting, mapping and interpretation of the data [75].

Ethical approval was granted by Flinders University Social and Behavioural Ethics Committee (project number 7101) prior to data collection. Consent processes involved information sharing via written format and verbal follow-up with paid interpreters provided by the researchers when needed. As the sampling did not rely on access to services by researchers or participants, anonymity was reaffirmed. Conducting interviews in homes and with interpreters approved by participants also contributed to an equitable data collection process that placed participants in positions of relative authority over their shared stories.

Rural settlement policy in Australia is such that groups of people from similar cultural communities tend to move to particular towns (2, 24). In this study, participants had backgrounds in two regionally defined cultural communities. Of the participants from African communities (*n* = 23), the majority identified Congolese heritage but there was variance in both heritage and migration journeys represented. Participants had lived in a range of places during refugee journeys in central and eastern sub-Saharan Africa, such as rural settings, refugee camps, and/or cities depending on individual circumstances. Herein, we refer to people from these communities as ‘African’ in order to maintain anonymity of participants whose specific cultural heritage may identify them to readers familiar with the town. Participants from South-East Asia (*n* = 21) originated from Burma/Myanmar, and prior to Australian migration most had lived in rural settings, often for many years or decades in refugee camps. Many of the participants from Burmese communities had similar experiences to one another prior to resettlement. Here, we use the term ‘Burmese’ as most of our participants had lived through the time when the country now known as Myanmar was called Burma. We recognise that there has been significant debate around the use of these names both in that region and globally in relation to the military junta in 1989 [76]. We do not differentiate between ethnic heritage of participants from this region in order to maintain anonymity. Participants represented a wide age range of 18–68 years, and around half were 25–50 years old. Around half of the group had lived in Australia for less than five years and the other half between five and ten years. Likewise, there were almost equal numbers of women and men represented.

All names used herein are pseudonyms.

## 3. Results

We report the findings of this study using excerpts from the data to show how the social ties that participants developed were helpful in enabling access to resources for health. These included emotional resources, such as a sense of belonging, as well as practical resources, such as food, housing, or accessing services. We first establish the formation of social ties, the resources gained through these ties, and the barriers and challenges to creating and maintaining ties. We then discuss employment in healthcare for people from the participants’ communities.

### 3.1. Social Capital and Access to Resources in Rural Refugee Communities

#### 3.1.1. Formation of Social Ties

Arriving in a rural town to begin a new life with limited services and few or no social networks was noted as challenging, as African refugee Etienne shared through an interpreter: “*when arriving in new area you are just a new person, you don’t know anything*”.

Community members often worked together to support newcomers, and cultural and linguistic elements of these ties were a particular strength. Henri noted the importance of community connections and how people arriving could draw on the knowledge and connections of people who settled in earlier:


*…now they are lucky because when they come we already face all the problem and we are able even to mentor them ‘if you want to do this, this is the way. If you want to do this, this is the way’ so that’s that.*
(Henri, Africa, man)

Myint Zaw Zaw likewise described how these connections and the cultural communities developed over time, as communities grew in size and the number of years of settlement:


*I mean because many family came after that and then [the service] aren’t able to look after all the families, that’s why [they] talk to the family who arrive first here …Like if my brother or my mum, they came after us, we have to look after them, take them to the hospital or where they need to go.*
(Myint Zaw Zaw, Burma, man)

Faith settings were key sites of bonding and bridging network development. Henri (Africa, man), described looking for and finding a church and the positive impact: “*there was no problem because when we came we looked for the church. We went there, then integration was easy*”. Once the initial phases of settlement merged into longer-term life in the town, bonding and bridging ties developed through faith communities remained important for participants’ sense of belonging. Participants spoke highly of cultural community bonding ties alongside faith, as Kampata described:


*I have one family here which I’m attached too much to them because of the spirituality. He’s someone I trust among all [people] instead of others…Because of the faith and he knows from where I am and I know him and he accept me as I am.*
(Kampata, Africa, woman)

Participants mentioned how important understanding language was in connecting to their faith, and there were churches that had been started by members of the African and Burmese communities. Although bonding cultural community networks often overlapped significantly with faith communities, bridging ties with people from culturally dissimilar communities were also made possible via the church. Some participants mentioned meeting ‘Australian’ people via the church as in the case of Kasambayi, “*Concerning Australian community… other relationship is through the church. With church we do create so many relationships*” (Africa, woman). Aeindra Thet Khine Hsu also explained:


*[The community] go and pray there, like in afternoon church, for their own community only...the church is really helpful for them, for all the community to connect…they do also go to Sunday church in the morning as well where the Australian people go to church, so they also go as well.*
(Aeindra Thet Khine, Burma, woman, via interpreter)

Younger participants discussed the bonding ties made in education settings. Sandar Thiri Cho described how initial bonding ties with friends from her cultural community at school soon spread out to include students from the broader community:


*…at first there was only us and there’s only me, I don’t have a friend, like my cultural friends, but then I think one month later like there’s a new family, like five new families coming, and then...we make friends very quickly and when we go to school we do everything together. But we also have like friends, like English friends, and we’re all friends together.*
(Sandar Thiri Cho, Burma, girl)

School settings were also places where parents/caregivers were in contact with other adults from outside of their cultural community, offering opportunities to develop bridging ties, such as for Kasambayi:


*Concerning Australian community, how we do meet is through our kids because they are schooling together so through there they can meet friends and then those friends can come to us when there is some kind of invite, you can invite their parents and then we know each other.*
(Kasambayi, Africa, woman, via interpreter)

Participants indicated a strong desire and sense of duty to connect with new arrivals and provide practical supports. Participants noted that it was important for settlement support services to connect with existing community members and let them know when new people were arriving so that they were able to offer support right from the beginning. For example, Gabi suggested that she could draw on her cultural and linguistic expertise to support new people arriving and settling in. However, she also noted that services were not always connecting new arrivals to the existing community:


*You are new. You don’t know what to do…you are not speaking English. You don’t know even [the town] well. What are you going to do? You have to remain at home and do nothing. But if [settlement services] connect, always connect the new arrival to the community… I’m in the better position to explain in my own language so they can understand better how to live here but they don’t do that.*
(Gabi, Africa, female)

#### 3.1.2. Resources Gained

Participants spoke of bonding ties within cultural communities, bonding and bridging ties in faith and education settings, and access to resources, including a sense of belonging, and important SDH, housing and employment.

Although participants accessed some of the specialised supports available to them upon arrival via the humanitarian program, the service centers in the towns were quite small, often with only one or two members of staff, as such supports provided as well as information about the available supports through these services were limited. Participants in this study spoke of how they worked to support newer arrivals who did not know how to gain access to resources important for their health and were not appropriately supported to access basic health services, Gabi noted:


*I was passing [the hospital], then I saw someone. He was struggling; he don’t know where to [go]. Then I stop, I say ‘Hello. Are you new?’ He say ‘yes’. ‘Do you need a lift?’ He said ‘yes’. ‘Where are you going?’…he said ‘I don’t know. They left me here. I don’t know where to go, please help me. I need to go to the clinic. I have an appointment. After that I have to go to see the people for blood test. Then I have an appointment in the afternoon. Then I helped him. Then I show him the shortcut ‘from your house you have to go like that’.*
(Gabi, Africa, woman)

Access to services important to health were facilitated by the process of initial welcome for new arrivals to the town, and some participants spoke of their linking ties to settlement services that enabled such community welcome. Keza (*Africa, woman*) talked about how a service provider kept ‘*communication with the community so the community will be aware that some people are coming*’ so that an organised welcome could be provided. While this was not always the case, as noted above in relation to Gabi, this is a key example of how service provision can be delivered effectively by tapping into African and Burmese community relationships. Emmanuel described benefits of this process:


*It’s like when we get information … like ‘there are some African people are coming’ people just organise themselves. They can go and welcome them from the airport, bring them home. They can prepare everything because they know all the people that… so they have to provide food, really food for Africans.*
(Emmanuel, Africa, man)

Moreover, socialising and building friendships and bonding ties within cultural communities in the town enabled a sense of belonging to develop, a resource not only for those newly arrived but also for those who had lived in the town for some years, as Myint Htet Yaza said:


*It’s good because, you know, a lot of the community here already settled here so a lot of them are settled here and they are all socialising, their own community*
(man, Burma, via interpreter)

Bonding and bridging ties through faith communities not only provided a sense of belonging but also access to practical resources such as getting the electricity connected, as Celeste shared:

*For the first two or three days we had no electricity so we had to sleep in darkness. People from [redacted] our fellow church member…they had come in with this heater that you use when you are painting. They helped us with that and they give us candles, torches, so after that they just told us* [redacted] *they called the electricity people.*(Celeste, Africa, woman)

In terms of housing, a key SDH, participants spoke of varying relationships with landlords or realtors. For example, Sydney described building trust with his landlord (a long-term resident of the town) who was happy to act a referee for subsequent rental applications. Sydney described how this bridging tie was able to be transferred or shared with newer arrivals from similar cultural communities and the importance of this relationship in light of previous experiences of discrimination and racism in the housing market experienced by newcomers:


*… we meet with… [a] real estate agent… so she was like our referee and she refer to get another house… Then when people, the new arrival, they want to apply we can refer them to our [real estate] agency because they see us and they know now how we are taking care of their house…*
(Sydney, Africa, man)

The Burmese community, whose members had been settling in the town over a longer period, compared to the African community, had also developed these kinds of reputations and bridging ties with real estate agents, as Thuta Myat shared: “*when you go and apply for one house it’s very easy if you are from Burmese background because they love [Burmese tenants]*”.

When finding housing was difficult or the housing itself was unsuitable, participants spoke of people drawing on their communities for support in other ways. Etienne, from Africa, described how his initial accommodation lacked adequate cooking facilities, and how he was helped by the African community who brought him food and gave access to resources for cooking. Etienne shared his sense of gratitude and how being shown care by his cultural community had a positive effect on his wellbeing:


*It’s only people in the community that were helping him, to go with him… He was really happy to find some people that did care about him when he had that matter and he was really, really affected and it was really impressed according to him because in time of having that difficult time people came to him and then helped him.*
(Etienne, Africa, man, via interpreter)

Some participants spoke of learning about job opportunities through their cultural communities, or of building on the hard working reputation of the community in the town when seeking work, even though overall unemployment rates were known to be high in the refugee groups (23; 24). For example. Thuta Myat spoke of how these types of networks provided opportunities for community members to link up with employers:


*Especially the aged care it’s easy because…our community…there are two or three [Burmese community members] that work in aged care…You know, they work there and then they work very hard and that’s why their boss is very happy with their work. So then I went there…I asked them if I could work here after I finished my certificate and they say yes.*
(Burma, man)

Another crucial SDH is access to healthcare, and interviews showed that bonding ties between members of cultural communities were key to people from the Burmese and African communities accessing healthcare. A major issue was the language barrier, and many participants detailed how their community facilitated access for people trying to engage with healthcare providers. As Myint Zaw Zaw explained, although sometimes (but not always) services provided interpreters, often the available dialect was not a good match and family members or members of the community were required to assist:


*…when they came here…they need to go to the hospital they need an interpreter. Also mostly we can’t get interpreter all the time. Sometimes yes, sometimes no…They call their family, one of the family who can speak a little bit English. Yeah, we do that this way...If the family not available they ask for help from the other family who can speak a little English, then they can[redacted] they help and go together and help them*
(Myint Zaw Zaw, Burma, male)

#### 3.1.3. Barriers and Challenges

Bonding ties within cultural communities were often facilitators of key SDH, namely, access to housing, employment and healthcare, with some limited bridging ties also contributing. However, barriers to developing bridging ties were also noted, as well as difficulties accessing health services.

Participants reported experiencing poor understanding in the broader Australian community of cultural nuances and conflation of ethnic identities, with some people assuming, for example, that all local people from Africa were Congolese, or all people from Burma have the same background and language, as Anton described:


*All black, they called African, they called [redacted] not even African, they call them Congolese. Even if you are Rwandese you are Congolese. If you are Sudanese you are Congolese. It’s that community called us [redacted] they call us Congolese, Congolese, Congolese. They don’t know this is Rwandese, this is Congolese, this is…*
(Anton, Africa, man)

The effect of this lack of cultural awareness acted as a barrier to developing a broader sense of belonging and connection in the town. Developing a sense of belonging and connection through bridging ties was similarly hampered by incongruent socialising practices. Socialising for participants was informed by collectivist cultural norms, and was different to the socialising practices of majority Australian town residents, making it difficult for participants to build bridging ties, as Myint Htet Yaza and Etienne noted:


*Social life with Australian community is a big challenge because the way they socialise is a bit different… in my country in the village we all know each other. I’m living here… we have neighbour but for seven years we not become friends. They never come to visit me and also I haven’t gone just to go and visit them because of maybe the culture or custom; I don’t know.*
(Myint Htet Yaza, Burma, man, via interpreter)


*It’s very hard because…where we are coming from in Africa, when we are living in the area like this every time, like weekend, we can just entertain together, like neighbors, we can just entertain. We can share dream, we can sit together and then trying to share some stories, but it’s not the same as here in Australia… because they’re always being just locked in their houses.*
(Etienne, Africa, man, via interpreter)

In terms of barriers to accessing healthcare, finding a way to locate and access services was a challenge for some. In some cases, participants discussed the difficulties getting the right healthcare in a small town and issues with travelling to larger cities to see specialists, which is a common challenge for all people living in rural areas. This was difficult for participants who had limited English, did not have a driver’s license or car and were living in a place that made it also expensive to travel and to find a place to stay in the city if needed. Maung Zeyar Myint from Burma put it this way:


*He have issue with getting to the hospital, like in Adelaide, because he often had to go to Adelaide to do health check up on him, but he sometimes struggled with that [because] he can’t communicate so he can’t ask for help to get on the way sort of thing.*
(Maung Zeyar Myint, Burma, man, via interpreter)

As mentioned above, language barriers presented challenges accessing a range of SDH, leading to people drawing on their cultural community ties for help. Although this was a community strength, participants noted some of the issues that arose due to inadequate provision of interpreting services by healthcare providers. For example, Haymar Aeindra (Burma, woman) spoke of issues understanding how Medicare (the government medical scheme that subsidizes healthcare) worked, which led to her not receiving the rebates she was entitled to. She also talked of medication errors due to insufficient interpreting. Moreover, Aeindra Thet Khine Hsu (Burma, woman) spoke of her undiagnosed pain issues and Myint Htet Yaza (Burma, man) explained that the dialect of interpreters was really important and that healthcare services did not always get the right supports:


*…[sometimes] the interpreter on the phone is from a different area where they speak [redacted] it’s the same language but it’s different pronunciation and sometimes could be mistake, like [redacted] and then sometimes the doctor can give them out different medication that will affect them.*
(Myint Htet Yaza, Burma, man, via interpreter)

These language barriers were also barriers to access in the first instance, in so far as communication between healthcare providers and refugee community members could not always be deciphered, leading to confusion and worry, as Thi Marlar Khine Yadanar explained:


*…a challenge to her is communicate to someone, like if she has an appointment and she receive a text from the daughter or from someone else she receive that message but she’s not 100 percent sure about this so she go to the hospital and ask about the message that she receive and ‘is it about my appointment or is it about’[redacted] something like that.*
(Thi Marlar Khine Yadanar, Burma, woman, via interpreter)

Sydney, from Africa, also spoke of how he recognised barriers to access, and that he felt restricted in his ability to improve his circumstances. He mentioned feeling a lack of power to have his concerns and needs heard and responded to at a systems level. Although he felt he could share these feelings within his bonded network, he did not feel that it was possible to convey these issues through bridging or linking ties:


*We talk to our friends [about] the challenge which we are facing, but we don’t have that capability or ability, power, to fight with that things because, you know, when you are a new arrival you can’t [redacted] you just complain by yourself and there is no help…*
(Sydney, Africa, man)

Although the previous sections detailed some of the strengths of social networks within cultural communities in rural resettlement contexts, in terms of vital services (i.e., healthcare, housing, employment), reliance on cultural communities may not be the best support available, for example, in emergencies as Thuta Myat discussed:


*Because their English barrier and then they can’t really go and ask for help…In case for example if their house on fire the first thing—we told them like to contact triple zero for fire so the thing is most of them, they will just contact their link family so, yes, that’s the problem.*
(Thuta Myat, Burma, man)

Some participants with either African or Burmese heritage (whom we do not quote for confidentiality) spoke of exclusions from what might be considered potential cultural communities due to their own ethnic heritage and identity. In this paper, we focus on the strengths of cultural community wealth and social capital; however, we note that these strengths do not serve every person settling after a refugee experience, and this is an important exception that must be recognised. Alternative supports for individuals facing these barriers must be considered.

### 3.2. Employment in Healthcare Roles

Study findings showed that there were many people within refugee communities motivated to join the healthcare workforce. Some of these participants were highly skilled health professionals whose qualifications and work experience from overseas were not recognised in Australian systems. These challenges were exemplified by the experiences of one participant who was a qualified medical doctor who detailed considerable barriers to getting qualifications recognised in Australia. Subsequently, after gaining multiple Australian qualifications to work in healthcare, he was still unable to gain employment.


*I worked almost 15 years [as a doctor…As soon as I arrived I] tried to register, tried to get my qualifications recognized. It was too hard, expensive and then too much formalities. I didn’t have finances to afford that. Then I opted to do nursing course, which I did as a post registration course, two years…I finished my bachelor degree in nursing. Now they couldn’t register me because they said ‘you are coming from overseas and then you cannot be registered’… so I opted to do another course. I went into disability, Master in Disability.*
(Henri, Africa, man)

Even with the Australian Bachelor’s and Master’s degrees, and extensive and significant overseas training and experience, Henri struggled to gain registration and employment. After striving to gain work commensurate with his skills and qualifications, he applied for supporting roles in healthcare, but was told his qualifications were not the right kind. He then thought that perhaps he could just volunteer to use his skills in some way, but even that he found difficult. Henri described these persistent barriers as “*very distressing, very depressing*” leading him to consider that “*it’s better I sit at home*”. (Henri, Africa, man)

Notably, Henri’s challenges meant that his skills and expertise were likely to go unutilised in the town. He was very keen to find employment in this field and was ready to migrate onwards from the town if necessary with his family, as he said, ‘*anytime if I find a job anywhere I won’t hesitate to go; everyone will just go*’.

There were other participants who reported a range of qualifications and experience in healthcare roles pre-migration as well as several who had plans to work in this field in the future after gaining Australian tertiary qualifications. Others had Australian certificate qualifications and were working in caring roles in disability or aged care. Some participants spoke of how members of their community were linked in with employment opportunities through social capital in the form of community reputation.

As detailed in earlier sections, some people in the town did not have a good understanding of people’s diverse backgrounds and experiences, and many participants detailed experiences of racism and discrimination in daily life and at work. As Dimena explained, she felt that service providers were not meeting her needs due to racist attitudes:


*It’s just like some people just don’t understand why Africans, they’re here and some, when you go ask for service they don’t give you to you… I feel because maybe I’m African…. For myself, sometimes I deal with the people and just ignore because when I—even sometimes I don’t give up, I just keep trying, trying. If I try one service and then they say no I still try another service because the people, they are not the same. There’s some still accepting African as they are people and there’s some people that don’t understand.*
(Dimena, Africa, woman)

Participants also spoke of experiencing racism in the health workplace, such as Tatiana, who described being allocated tasks by her Australian peer as the nurse manager routinely chose not to communicate with her directly:


*…there are some kind of racism there at the workplace because -- I can explain: you can be working with two carers but once it’s time to explain something, if it’s an Australian with a nurse there, they talk to the Australian carer and then that one will come to you and explain things, what you are supposed to hear… Consider that other carer, like your boss on top of you […] I did talk to the nurse straightforward ‘what you are doing is not fair. Why would you talk to her instead of me? We are all carers’…[and the nurse said] ‘Oh, no, there is nothing wrong’. I said’ yes, there is’. Then she say ‘okay, next time I’ll never do that again’ because she know that maybe I could go further and report.*
(Tatiana, Africa, woman)

Tatiana went further to say that she also experienced racism from residents in the care home she worked in:


*Maybe people here in [the town], they don’t used yet to Africans I think; I don’t know. Even in aged care where I am working, some residents, they don’t like black skin.*
(Tatiana, Africa, woman)

## 4. Discussion

This study illustrated how social capital manifested within refugee background communities resettled in a rural town, contributing to health via SDH, including through improving access to housing, employment, and healthcare, as well as a sense of belonging and integration more generally. The study also showed the potential for new arrivals to fill crucial healthcare employment gaps in rural settings. However, the findings also highlighted the range of barriers for people from refugee backgrounds in building social capital in rural settings as well as impediments to employment in the health sector, which are important to consider for resettlement and health planning in rural areas.

### 4.1. Social Capital: Social Networks and Connection to Resources

As people from refugee or migrant communities settle in rural locales, they can draw on multiple ties with people with similar experiences, as was the case in our study. While there was some evidence of social bridges with members of the majority community through church and school, resulting in access to resources associated with friendship (sense of belonging and welcome), participants of this study predominantly reported on co-ethnic bonding ties. It is generally assumed that over time these networks can expand to include more diverse social ties, which may offer increased access to social capital. However, various factors, such as different cultural practices or countries’ dominant language, may hamper these processes [77]. Although positive resettlement outcomes have been reported over time, there are still systemic barriers to intercultural connections [36] and challenges to be faced when diverse cultural belief systems interact [23]. This can be true when different communities from refugee or migrant communities interact, as well as when minority and majority communities interact.

In the data, social ties assisted participants’ feelings of belonging and connection within their cultural communities and also linked people to other resources for health, a process of social connection which has been shown in other settlement contexts [78,79]. Connections to pre-migration ethnic identities, languages and memories combine with post-migration experiences to affect people’s feelings of belonging [80]. Physical connections through daily conversations in home languages, through religious practices or through shared familiar foods provide links to culture and a sense of ongoing shared community and place [81]. These feelings of belonging are important to people’s decisions to settle and build a life in these rural locations, as well as being key SDH.

The findings lend support to the notion that the conceptualisation of bonding and bridging ties as a simple dichotomy may benefit from development of the concept of a spectrum from bonding to bridging [52]. Simplistic conceptualisations of bonding and bridging capital can overly dichotomize cultural communities and majority cultures, which may be both inaccurate and detrimental [78]. However, in the rural Australian resettlement context, these dichotomies may be more pronounced as a result of the historic cultural and linguistic homogeneity of these places. In this context, our data suggest that building on the strength of social capital in cultural communities through the formation of bridging ties with members of majority communities can have positive outcomes.

### 4.2. Social Capital and Social Determinants of Health

The data indicated how social capital had an impact on health as an SDH and through other key SDH important to integration. For example, in Australia, English language proficiency remains crucial for public interactions or service access such as education, employment and housing, as well as health services [82]. The skills of key multilingual people can be a strength of collectivist communities, as shown in this study. However, there can be an overreliance on cultural community support by service providers [83]. A number of participants spoke of interpreting for family or friends because the service they were accessing did not supply effective interpreters. Some people were even made to pay for their interpreting when accessing key services such as healthcare, which not only increased risk for people who may have communication difficulties, but also because the cost may deter people from seeking healthcare (or other key services) in the first place. Settlement service providers may be able to support knowledge sharing via community networks about rights and responsibilities in the Australian context. It is important that mechanisms are put in place so that service providers do not rely on informal community interpreting that takes advantage of individuals and communities. If the process of connecting to community wealth [57] is performed with these risks in mind, and adequate processes are put in place to reduce the burden on community members where relevant, this could be an excellent way to draw on social capital and connect people to services.

In terms of access to healthcare, findings from this study indicate that healthcare systems can improve through outreach to these communities to link with networks that are functioning well. Engaging with communities in an active way could present an opportunity for healthcare provision that builds the trust of healthcare services, which is particularly crucial for resettled refugee communities that may have experienced challenges trusting systems and whose knowledge of settlement countries systems is developing [52,72], as well as further supporting the development of social capital. Employment of people from these communities would provide people with access to resources for themselves as well as improving social ties between healthcare services and people in these communities.

### 4.3. Local Health Workforces

There is opportunity for healthcare services to contribute to the SDH through employment of people from refugee communities. There is also considerable opportunity for people from refugee communities to contribute to better healthcare for people in their communities and to the wider population of the town. However, stereotyping and racism from the broader community showed a lack of general understanding of these communities by some, which could flow into healthcare provision and as such issues relating to racism need to be addressed. Cultural safety for consumers of healthcare has been noted as central to effective healthcare provision, but is also important for people from minority groups who are employed in healthcare settings [84].

Challenges finding employment, noted in this study, were juxtaposed for some with a high level of pre-migration education and work experience, often in healthcare roles, such as doctors and nurses. For example, Henri’s experiences showed how the healthcare systems in the town failed to draw on his significant skills and expertise. This was a missed opportunity to not only contribute to the health and wellbeing of Henri and his family, but also to the health and wellbeing of the whole town population that could have benefited from his expertise. Although his story was a particular example of these issues, it was not an isolated experience and several participants spoke of their training and experience working in healthcare that were not recognised in Australia. Several participants had also studied or were currently enrolled to study in Australian vocational and tertiary qualifications in healthcare, such as carer/support roles or physiotherapy. Hence, the study identified that there is a lack of employment opportunities for people within the two refugee communities in the rural town, and that many of these people were keen to work in healthcare. Participants aspired to stay in the town if they could find work, which could be a source of healthcare workforce, which is a persistent issue in rural locations in general. In combination with the lack of healthcare workforce in Australian rural and regional areas, this provides a win-win opportunity for workers and for healthcare services. As social connectedness of healthcare workers is an important element for attraction and retention in rural locations [85].

Rather than only emphasising the deficits and challenges for rural health, the facilitators and enablers of health and healthcare in these locations are important to highlight. Rural communities have strengths that can be drawn on to provide innovative solutions to health issues, and connections between healthcare services and communities in these locations may present opportunity for such solutions [86]. Provision of healthcare to rural refugee communities could benefit from healthcare workers who have similar cultural and linguistic backgrounds to those they provide care for. As detailed in this article, members of resettled refugee communities are well placed to draw on their social capital to both facilitate access for their community as consumers, as well as to provide social connections for themselves as longer term healthcare workforce members.

### 4.4. Limitations and Strengths

This study was able to draw on the experiences of a range of people from refugee backgrounds in a rural setting and to explore in depth their social ties and resources and relevance for health. However, as this was a qualitative study that used a snowball sampling to recruit participants, we might have recruited participants from the same networks of community members who knew each other only. As a consequence, we may have missed different perspectives or experiences of social capital and resettling in a rural setting.

## 5. Conclusions

This study engaged social capital and social determinants of health frameworks to understand the ways in which resources for health were accessed by refugees from two communities in a rural setting in Australia. In our study, we have shown the importance of social capital among individuals and within these communities. Specifically, we have shown that these communities were able to access resources associated with a sense of belonging and welcome and to access information about housing, employment, and healthcare mainly from bonding ties within their own cultural community. There was some evidence of resources gained through bridging ties with members of the broader community through church and school settings; however, strengthening social connection between refugee and migrant arrivals and the majority population is likely to be beneficial for all.

Policies for healthcare in these locations that draw on these strengths and connections may provide opportunities to improve health outcomes through enabling access to healthcare, as well as by contributing to the SDH by providing employment opportunities for people from diverse refugee communities living in rural areas. Not only can employment of these individuals add to their own health and that of their families via the SDH, but their presence in healthcare settings can contribute bridging and linking capital to the strong bonding capital with their communities, and further enable access to healthcare for their communities in that way.

Culturally distinct ethnic communities may be identifiable, but they also have internal diversity [87]. It is important to acknowledge heterogeneity within a group [78,87,88,89]. Service provision that builds on the strengths of cultural communities’ social capital must be developed with nuances of individuals and groups in mind. Importantly, while collectivist approaches can be a helpful connector to resources for some, for those who experience exclusion from their community, collectivism can also presents barriers to resources [90].

Likewise, while the strengths of the two communities in our study are a clear starting point for positive policies that affect engagement and access, there is a potential for service providers to rely on these communities to make their own way and provide supported access points for each other. The responsibility for service provision can be shared, but the duty of institutions and governments remains to provide equitable access for all. We note that although the refugee communities in our study drew on their social ties to access resources for health, these strengths sit in juxtaposition to some of the restrictions and barriers to health and cannot make up for other insufficiencies in infrastructure, workforce, funding or governance [74].

## Data Availability

Because of ethical considerations, the data presented in this study are not publicly available.

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
