# Peer review of "Social Capital and Rural Health for Refugee Communities in Australia"

_ijerph, 2023, doi:10.3390/ijerph20032378_

Round 1

Reviewer 1 Report

Title:

1.    Need to revise. Replace “Communities from refugee background” with “refugee communities”.

Abstract:

2.    Research questions and objectives should be clearly stated in the abstract.

3.    Research approach, data collection methods, and analysis methods should be added.

4.    What are the core findings of this research- should be concisely added at the end of the abstract.

5.    Remove the jargon, and make it precise and informative.

Keywords:

6.    Why “cultural wealth” is separate keyword while having “social capital”?

7.    “Integration” seems not an appropriate keyword.

Introduction:

8.    Establish “why this research is necessary” at the beginning, and increase more numeric evidence.

9.    Why SDH lens is used- establish scholarly, comparing other similar lenses.

10.  L-42-44, discuss the number of participants and research approaches in the methodology section.

11.  A scenario of social capital and health regarding existing rural Australian communities is necessary to get an idea of what differs with refugees.

12.  The theoretical relevance of studying “social capital” and “health” together is missing. Prepare a conceptual framework based on previous theoretical frameworks.

13.  L-168-171, discuss the methodology in the relevant section, do not repeat any information.

14.  The overall focus of this study is unclear. Social capital and health are both relevant study topics for rural study and refugee resettlement studies. However, none of them are made as the main topic, or no strong linkage is established. There are a lot more issues that could be addressed, why it did not.

Materials and Methods:

15.  Data collection methods and sample details are well described.

16.  NVivo as a tool is used as mentioned, but how it is helping in the analysis is still unclear.

17.  Developing an analysis framework would be helpful. Align the main research question, objectives, data collection methods, tools, data analysis methods, and expected outcome.

18.  As it is a case study research on the two distinctive regions, a Location map is necessary.

Results:

19.  Different analysis frameworks are mentioned in the methodology section, however, no evidence is presented.

20.  L-237, [N]ow- check this word.

21.  Quotations are unnecessarily lengthy, interpretations are shallow.

22.  More original and in-depth findings are expected on both “social capital” and “health” studies from such a qualitative research

Discussion:

23.  The narrative approach in the Discussion section is appreciated, trying to relate to existing knowledge. Themetical brief of the findings is necessary.

24.  Reduce unnecessary text, and make it concise and aligned with the research objectives.

25.  The discussion seems biased as it is unclear- how the research outcome is drawn.

Conclusion :

26.  An overall summary should be drawn reflecting all the key steps.

27.  As it is qualitative research, there should be a brief “limitation” part in the conclusion section.

28.  How the knowledge of this research outcome would help and be useful for these communities should be addressed.

Reference:

29.  Cited references are relevant to the study and from recent times.

Author Response

Dear editors, we have responded to reviewers comments point by point

Reviewer 2 Report

The manuscript "Social capital and rural health for communities from refugee background in Australia" presents a practical reality of social capital impact on  refugees and asylum seekers  in host communities. However, authors talk about key social capital theorists but fail to provide a detailed theoretical frame work that underpins the study.  You have discussed Bourdieu's conception of social capital in the passing and I  suggest you add a short paragraph detailing how social capital and networks  at both individual and community levels are sustained.  There is an absence of theory in your discussion section.  Delete the last sentence at 1.3 or shift it to methods section.

Author Response

Response: Thank you for comments.  We have added a brief section on social capital and how this is sustained at individual and community level. We have shifted the last sentence of 1.3 to methods section

Reviewer 3 Report

Thank you for giving me an opportunity to review a manuscript entitled “Social capital and rural health for communities from refugee backgrounds in Australia”, Main focus on Australia's migration programs benefit greatly from the resettlement of refugees, and recent policy directions have prioritized rural resettlement. As a result, the cultural diversity of the people living in a number of rural Australian towns has greatly increased. Integration of newcomers into closed social networks and access to the resources needed for a healthy existence in the settlement may be difficult. Still, there are benefits for both newcomers and receiving communities that result from successful integration. The findings of in-depth interviews with 44 participants from Africa and South-East Asia who moved to a rural South Australian town are discussed in this study using a social capital framework. The connections participants made within their cultural communities, which reflected collectivist cultural values and gave them access to resources for health, including health services, were a key source of inspiration. A few participants were also pursuing careers in the healthcare sector. Rural towns' populations of people with refugee backgrounds are expanding due to both the steady influx of newcomers and the growth of the settled populations' families and communities. There is potential for services to tap into community networks to help serve these growing communities in these locations through effective health service provision. The advantages and difficulties of fusing social networks with healthcare delivery are discussed in this paper. Although the manuscript is well-written and organized. However, Overall, the manuscript is good. The Paper can be accepted after a minor revision: Some comments are given below. It is better.

1.      I advise the author to make minor changes in order to eliminate grammar roughness and to improve the body text imperfection.

2.      Material and Methods section must be re-written. It is necessary to explain better the method of the research, the hypothesis, etc. In the research is there a survey?

3.      What is the theoretical contribution of this research? can you compare your findings with the previous studies.?

4.      Conclusions sections must be re-written. This section must reflect the conclusions of the research.

5.      As a supplementary file. Data analysis I will suggest to rename this section as result analysis likewise, renaming result discussion to only discussion and subdivide.

6.      The relevance of the conclusion analysed about employment opportunities. I'm particularly in doubt regarding employment opportunities which apparently does not provide any Literature review; It’s Lack of support or added information for the article. besides being from an external source.

Good Luck to the authors 

Author Response

We have responded to reviewers comments in the attached file

Round 2

Reviewer 1 Report

Revisions and responses are impressive. I would suggest going for a minor spelling and grammar check (ex: P.15, L 757: " 5.1.1...." this is an error). If possible, improve the readability of the article.